# Human-Aware Collaborative Robots in the Wild: Coping with Uncertainty in Activity Recognition

**DOI:** 10.3390/s23073388

**Published:** 2023-03-23

**Authors:** Beril Yalçinkaya, Micael S. Couceiro, Salviano Pinto Soares, Antonio Valente

**Affiliations:** 1Ingeniarius, Ltd., R. Nossa Sra. Conceição 146, 4445-147 Alfena, Portugal; micael@ingeniarius.pt; 2Engineering Department, School of Sciences and Technology, University of Trás-os-Montes and Alto Douro (UTAD), Quinta de Prados, 5000-801 Vila Real, Portugal; salblues@utad.pt (S.P.S.); avalente@utad.pt (A.V.); 3Institute of Electronics and Informatics Engineering of Aveiro (IEETA), University of Aveiro, 3810-193 Aveiro, Portugal; 4Intelligent Systems Associate Laboratory (LASI), University of Aveiro, 3810-193 Aveiro, Portugal; 5INESC TEC, Campus da Faculdade de Engenharia da Universidade do Porto, Rua Dr. Roberto Frias, 4200-464 Porto, Portugal

**Keywords:** human activity recognition and modelling, deep learning, human-robot collaboration, fuzzy logic, finite state machine, long short—term memory

## Abstract

This study presents a novel approach to cope with the human behaviour uncertainty during Human-Robot Collaboration (HRC) in dynamic and unstructured environments, such as agriculture, forestry, and construction. These challenging tasks, which often require excessive time, labour and are hazardous for humans, provide ample room for improvement through collaboration with robots. However, the integration of humans in-the-loop raises open challenges due to the uncertainty that comes with the ambiguous nature of human behaviour. Such uncertainty makes it difficult to represent high-level human behaviour based on low-level sensory input data. The proposed Fuzzy State-Long Short-Term Memory (FS-LSTM) approach addresses this challenge by fuzzifying ambiguous sensory data and developing a combined activity recognition and sequence modelling system using state machines and the LSTM deep learning method. The evaluation process compares the traditional LSTM approach with raw sensory data inputs, a Fuzzy-LSTM approach with fuzzified inputs, and the proposed FS-LSTM approach. The results show that the use of fuzzified inputs significantly improves accuracy compared to traditional LSTM, and, while the fuzzy state machine approach provides similar results than the fuzzy one, it offers the added benefits of ensuring feasible transitions between activities with improved computational efficiency.

## 1. Introduction

### 1.1. Importance of Human Activity Recognition for Human-Robot Collaboration

For years, robots and humans have been separated in different workspaces, whether it be industrial or field applications. The reason for this separation is primarily for safety. Even though robots have been designed for specific tasks, in most cases, they are not aware of the environment and surrounding dynamic agents. As a result, these robots are often placed in cages or in a completely separate environment from human operators [1]. This separation has resulted in issues, such as low adaptability in different environments, costly setup, and limited flexibility, which do not align with the ideals of Industry 4.0, which demands fast production and efficiency.

To address these demands, Human-Robot Collaboration (HRC) has become a major trend in robotics in recent years. The goal is to improve efficiency and productivity by combining the benefits of humans’ critical thinking and empathy, with robots’ physical robustness in demanding and often dangerous conditions [2]. The idea is for humans and robots to work together towards a common goal. Research has demonstrated that interaction and collaboration between humans and robots are crucial factors in achieving ergonomic systems and enhancing the quality and efficiency of the production process [3].

Collaborative robots, also known as “co-bots”, have become increasingly prevalent in industrial settings in recent years. However, they have also been utilized in a variety of other domains. For example, in the healthcare field, researchers are developing robotic walkers [4], wheelchairs [5], and elderly care robots [6]. Collaborative robots have the potential to assist humans in heavy and dangerous tasks as well, such as construction and search-and-rescue [7]. They can also be used in a range of industries and even in smart home applications. These robots can come in various forms, such as manipulators [8] and fully humanoid robots [2].

However, incorporating humans into the process presents many challenges, primarily due to the unpredictable nature of human behaviour. This can lead to difficulties with robots adaptability and robustness in changing and uncertain situations and environments. In HRC systems, robots are expected to understand human activities and intentions and, at times, even predict future human behaviour in order to efficiently achieve the shared goal. This can be a difficult task due to the inherent uncertainty of human behaviour.

Significant research has been dedicated to understanding human behaviour patterns through Human Activity Recognition (HAR), which involves analysing various sensor data to identify and detect simple and complex human activities. HAR has been applied not only to domains related to human daily life, such as healthcare, smart home applications, and elderly assistance [9], but also in robotics solutions where HRC is foreseen, being critical for the robot to have awareness of human actions. Traditional machine learning methods, such as Bayesian networks [10], random forest [11] and support vector machines [12] have been used to understand human behaviours. In addition to understanding human behaviour, some researchers have focused on predicting the most likely sequence of human actions. Probabilistic methods, such as Hidden Markov Models (HMM) [13] have been proposed to understand and predict human activities. Finite State Machines (FSM) have also been used as a tool to model dynamic changes over time and, when combined with fuzzy logic, to even handle uncertainty from sensor data through the use of linguistic variables [14]. Recently, deep learning has emerged as a new trend, as it has the ability to learn and identify complex patterns among large datasets. The major difference between deep learning and the previously described approaches is that it offers multiple hidden layers that are capable of feature extraction and transformation, thus significantly reducing the workload of human designers and developers. As a result, deep learning has been used in various other domains as well, such as image classification [15], speech recognition [16] and so on, and several deep learning algorithms, such as convolutional neural networks (CNNs) [17] and recurrent neural networks (RNNs) [18], have been key to improve the accuracy and robustness of HAR systems.

While these methods have shown promise, dealing with human uncertainty remains a challenge. One of the main difficulties is the high variability of human behaviour across different contexts, as well as the noise in the sensor data, which makes it difficult to generalize from training data. This uncertainty problem has a negative impact on trust and safety, which are critical measurements for any HRC system [19]: if the robot is unable to understand or anticipate human intention, this may lead it to make wrong decisions and even cause accidents and injuries, which will affect both acceptability and trustworthiness. Several authors point out to a panoply of solutions to eliminate uncertainty by extracting more information and rapid processing. However, there is no clear plan established for a constrained computing system, as robots and other facilitators of HRC (e.g., wearable technologies) often have. Given the complexity and addressed challenges associated with uncertainty in human behaviours, further research is still required to fully understand and address this problem.

### 1.2. Research Question and Objectives

This paper proposes a HAR framework capable of coping with uncertainty in human behaviours, resulting in positive improvements in trust and safety for HRC tasks. To this end, this paper presents three key incremental developments:Enhance Long Short-Term Memory (LSTM) networks by incorporating fuzzy logic to model human uncertainty (Fuzzy-LSTM), building upon the work of [20]: the goal is to improve the performance of LSTM networks by incorporating fuzzy logic to model human uncertainty. In this method, features are extracted from sensor data, which may be uncertain due to the ambiguity of human behaviour or noise in the sensors. These features are then fuzzified using Tilt and Motion linguistic variables. This fuzzification step allows the model to handle uncertain data, making it more robust. The fuzzified features are then used as input to the LSTM network during training. The goal of this approach is to improve the accuracy of the LSTM network in handling uncertain sensor data.Further extend Fuzzy-LSTM representing the sequence of activities through finite-state machines (FSM), thus leading to the Fuzzy State LSTM (FS-LSTM): the goal of this method is to enhance the predictability of human activity sequences by combining the strengths of FSM and fuzzy logic in an LSTM-based model. In this approach, an LSTM network is trained for each state within the FSM. The output of the LSTM network is then used to determine the possible transitions between states.Estimate human uncertainty by aggregating predicted scores of the LSTM into a crisp output through defuzzification: this proposed method aims to estimate the uncertainty of the LSTM classifier’s predictions by converting the classification scores into a crisp value through defuzzification. The classification scores are first converted into a fuzzy set to represent the degree of uncertainty in the predictions. Then, the fuzzy set is transformed into a crisp value to indicate the certainty of the classifier’s predictions. This process allows for quantifying the uncertainty of the predictions, which is not only used within the FS-LSTM method to accept or reject transitions between states, but can also be useful in our future work in HRC, where certainty is important.

In addition to these three main contributions, a benchmark is presented to further investigate the impact of the proposed architecture, which compares the traditional LSTM and the incrementally developed novel architecture.

### 1.3. Organization of the Article

This article is structured as follows: In Section 2, a comprehensive review of relevant literature is provided. Section 3 outlines the use case and data collection through the developed simulator and a preliminary experimental study on generating synthetic data. The proposed method, including feature fuzzification and FSM learning with LSTM, is described in Section 4. At last, the results from the experimental studies are discussed in Section 5, followed by a description of future work and conclusions in Section 6.

## 2. Literature Review

HAR has gained significant attention for its ability to detect and identify human activities from sensor data [21]. The importance of HAR lies in its ability to handle the uncertainty that arises from the variability in human behaviour and ambiguity in activities. In robotics, understanding human behaviour and adapting accordingly is crucial for natural and safe interaction and collaboration with humans [22]. This literature review will examine the uncertainty problem in HAR and the methodologies used to address it, as well as the challenges and open research questions in the field.

Human uncertainty poses a significant challenge in HRC from various perspectives. One major aspect is the complex sequential decision-making required in dynamic environments during collaborative tasks, as discussed by Osman in her study [23] on complex dynamic control tasks. These tasks often require multiple decisions that have to accommodate many elements of the system to achieve a desired goal, which implies that there is a high degree of uncertainty introduced by humans regarding how they will behave in these changing conditions and environments. This makes it a difficult task for robots to predict and adapt. In addition to the dynamic nature of the environment and the complexity of collaborative tasks, the variability of human physical and cognitive abilities also contributes to their uncertainty. Human factors, such as fatigue, learning ability, and attentiveness can significantly impact a worker’s efficiency and accuracy and even cause errors or safety issues in HRC systems [24]. This is aligned with the work of Vuckovic et al. [25], that highlighted the importance of human subjectiveness in creating uncertainty in human behaviours. According to the authors, individuals judge a stimulus and adapt their decisions accordingly to their judgments. This implies that human subjectivity has an important role in introducing uncertainty in human behaviours, as it leads them to perceive and react to situations differently based on their own experiences. Another important aspect in which uncertainty plays a role is in building trust between humans and robots in collaborative tasks. Trust is a crucial element in HRC, as it allows humans to rely on robots to safely perform tasks together. According to Law and Scheutz [26], understanding human needs and intentions, and effectively responding to them, is key to building trust.

Other researchers have emphasized the significant impact of human uncertainty on proactive planning for HRC. According to Kwon et al. [27], proactive planning involves a robot’s ability to adapt to a dynamic environment by handling uncertainty. The authors note that the nature of the dynamic environment is not only affected by the robot’s actions but also by human activities, which have complex temporal relationships. The uncertainty in these activities must be considered during planning as they are not easily predictable due to the robot’s limited observation of the environment and the humans. Therefore, understanding and addressing uncertainty in collaborative tasks is essential for efficient planning in HRC.

Based on the literature reviewed, it is well understood that uncertainty introduced by humans poses a significant challenge in Human-Robot Collaboration (HRC). Therefore, a significant amount of research has been conducted in this field with the aim of mitigating the negative effects of uncertainty. These solutions mostly focus on the efficient and effective inference of human behaviour as a means of addressing uncertainty within the HRC context. One approach is the use of multimodal systems that combine different sensor types, such as video cameras, wearables, and even ambient sensors, such as infrared motion detectors. Video cameras are popular for HRC tasks, but they raise privacy concerns [28]. On the other hand, wearable sensors, such as inertial measurement units (IMUs), are widely used to cope with privacy and security concerns, but they also come with many challenges, such as limited representativeness of similar activities. Despite these challenges, wearable sensors are the most commonly used set of sensors in human activity monitoring. In [29], the authors proposed to use wearable sensors, such as accelerometers and gyroscopes worn at different positions on the human body, to capture activity data that are sampled at regular intervals to be used in HAR. Another study has been designing appropriate methodologies, such as utilizing data from individual accelerometers at the waist, which can identify basic daily activities, such as running, walking and lying down [30]. These works reported acceptable accuracy results for basic daily activities. However, they could not show good accuracy for more complex activities, such as transitions, e.g., standing up or sitting down. As said before, a way of improving such results would generally imply using a larger combination of sensors, although attaching many sensors to the human body is unfeasible and inconvenient for people’s daily activities.

HAR is often treated as a pattern recognition problem, and many works have initially adopted machine learning techniques to recognize activities. Support Vector Machine (SVM) [31] and Hidden Markov Model (HMM) [32] classifiers are among the most commonly used methods for activity recognition. For example, Azim et al. [33] used an SVM classifier with trajectory features for activity classification and achieved an overall accuracy of 94.90% for the KTH online database and 95.36% for the Weizmann dataset (http://www.wisdom.weizmann.ac.il/~vision/SpaceTimeActions.html (accessed on 2 February 2023)). Kellokumpu et al. [34] used HMM and affine invariant descriptors, achieving an overall accuracy of 83.00%. While these works rely on offline data, Yamato et al. [35] used real-time sequential images and mesh features along with HMM, achieving a 90% accuracy. However, these traditional machine learning methods often rely on carefully designed and heuristic feature extraction methods, such as time-frequency transformation, statistical approaches, and symbolic representation. They lack a universal or systematic approach for effectively distinguishing human activities, and they are prone to overfitting and may perform poorly on unseen data [36].

To overcome these drawbacks, ensemble classifiers have been proposed, which involve training multiple models and combining their predictions to make a final decision. The aim of ensemble classifiers is to improve the performance of the model by combining the strengths of multiple models and mitigating their weaknesses [37]. Random forest is a popular ensemble classifier that is computationally efficient and commonly used in various domains, such as text and image classification. Random forest works by training multiple decision trees and combining their predictions through a voting procedure. This method is effective in addressing overfitting issues and has been shown to enhance accuracy by combining the outcome of each different classifier [38].

Both traditional machine learning and ensemble classifiers methods for feature extraction in HAR heavily rely on human experience and domain knowledge. However, these may not be effective for more general environments and may result in a lower chance of building an efficient recognition system. Additionally, the features learned by these methods are shallow, such as statistical information, and can only be used for low-level activity identification, such as walking or running, making it hard to detect high-level or context-aware activities, such as cooking. In contrast, in real-life scenarios, activity data comes in a stream and requires robust online learning from static data, which is a limitation of many of these traditional methods [39]. Deep learning methods, on the other hand, have been successful in learning complex activities due to their ability to learn features directly from the raw data hierarchically by performing nonlinear transformations. The layer-by-layer structure of deep models allows learning from simple to abstract features. Advances in computer resources have made it possible to use deep models to learn features from complex data from single or multimodal sensory systems. It is worth highlighting that deep neural networks can be detached and flexibly composed into a unified network, allowing for the integration of various deep learning techniques, such as deep transfer learning, deep active learning, and deep attention mechanism. This enables the integration of various effective solutions that can improve the performance of the recognition system [36].

Popular deep learning techniques include deep neural networks (DNN), convolutional neural networks (CNN), recurrent neural networks (RNN), and long short-term memory (LSTM) networks [28]. DNN are a type of Artificial Neural Network (ANN) that are characterized by a larger number of hidden layers. In contrast to traditional ANN, which often have only a few hidden layers, DNN can learn from large datasets more effectively. Hammerla et al. [40] adopted a five-hidden-layer DNN to perform automatic feature learning and classification. Vepakomma et al. [41] fed extracted hand-engineered features obtained from the sensors into a DNN model. CNNs are a type of neural network that exploit three key concepts: sparse interactions, parameter sharing, and equivariant representations. CNN have presented successful results in HAR application by utilizing local dependency, which refers to the nearby signals in a time-series that are most likely correlated. CNN also have shown the ability to handle variations in pace or frequency [39]. Several studies, such as [42,43] have employed one-dimensional (1D) on the individual univariate time-series signals for temporal feature extraction. Conventional 1D CNN have a fixed kernel size, which limits their ability to discover signal fluctuations over different temporal ranges. To address this, Lee et al. [17] combined multiple CNN structures of different kernel sizes to obtain the temporal features from different time scales. Nevertheless, this approach would demand more computational resources as well. Various deep learning methods have been applied to temporal information including RNN. While traditional RNN cells suffer from vanishing gradient problems, LSTM, as a specific type of RNN, overcomes this issue. A sliding window is generally used to divide the raw data into individual pieces, which are then used to feed LSTM. In a typical LSTM-based temporal feature extraction, it is essential to carefully tune the hyper-parameters, such as the length and moving step of the sliding window. Some researchers adopted The Bidirectional LSTM (Bi-LSTM) structure for extracting temporal dynamics from both forward and backward directions in HAR [44]. On the other hand, Guan and Plötz have combined multiple LSTM networks in an ensemble approach and obtained superior results [45].

Another trend in HAR is combining different deep learning approaches by developing hybrid models to exploit their different aspects. For instance, Ordóñez and Roggen have combined CNN and LSTM for both local and global temporal feature extraction [46]. The idea is to exploit CNN’s ability to capture the spatial relationship, while LSTM can extract the temporal relationship. According to the reported results, CNN combined with LSTM outperforms CNN combined with dense layers. Differently, in [47], the authors presented a hybrid model for HAR which first identifies the abstract activity by using random forest to classify it as static and moving. For static activities the authors have used SVM, while for moving activities they have adopted 1D CNN. Even though the overall accuracy of the system was 97.71%, their system was evaluated over a dataset and has not been tested in real environments and/or in runtime.

Despite these models having shown significant accuracy in HAR, the uncertainty of the activities remains a challenge due to several reasons, such as noise in sensors and human factors. Several studies adopted different methodologies to investigate the degree of certainty, or uncertainty, of a given performed activity. One of the methods adopted was a dynamic Bayesian mixture model (DBMM), which is a type of ensemble probabilistic model that combines the likelihood of multiple classifiers into a single form by attaching different weights to each classifier. DBMM uses an uncertainty measure, such as the posterior probability, as a confidence level, which is updated during the online classification [48]. Therefore, the classifier with the highest confidence level is the outcome of the classification process. In [49], the authors presented an architecture that recognises seven different actions performed by athletes using a single-channel electromyography (EMG) combined with positional data obtained by benchmarking ANN, LSTM and DBMM. According to the results, ANN and LSTM models were not the most reliable choice to identify these actions due to the low number of trials in the dataset. On the other hand, DBMM led to better results, with 96.47% accuracy and 80.54% F1-score. Similarly, in [50], human daily activities were recognized by using DBMM. The authors proposed a set of spatio-temporal features, including geometrical, energy-based and domain frequency features to represent the different daily activities which were then fed into DBMM. The overall classification performance for DBMM and LSTM, in terms of precision and recall, was 86.63% and 85.01%, respectively.

Other studies have explored fuzzy-based architectures in HAR, which allows for the incorporation of uncertainty in the decision-making process. While traditional probabilistic models represent the likelihood of an event using crisp values, fuzzy-based models use fuzzy membership values to represent the degree of partial truth by providing semantic expressiveness through the use of linguistic variables to handle uncertain data. Karthigasri and Sornam [20] fuzzified the input features to be used in a fuzzy FSM (FFSM), which is a methodology used to model dynamic sequences of events. The reported results of the approach outperformed decision trees, K-nearest neighbors, SVM, Gaussian naïve Bayes and quadratic discriminant analysis. Mohmed et al. [14] proposed a HAR architecture using data obtained from low-level sensory devices by enhancing FFSM with deep learning methods, namely LSTM and CNN. While both models have shown high scores of accuracy, the CNN-FFSM model showed more robust and reliable performance when applied to a larger dataset, while LSTM-FFSM outperformed CNN-FFSM for simple scenarios with a short period of a dataset. Despite the paper presenting promising results for HAR, the methodology is presented in a high-level manner, lacking relevant technical and scientific details, which makes it impossible for the reader to understand and fully asses its reproducibility.

In conclusion, the literature reviewed in this study highlights the importance for robots to understand human activities and cope with uncertainty in HRC applications. To this end, a variety of studies have been conducted in this field to understand human behaviour by exploring HAR architectures. However, it is clear that there is still a need for further research in this area in order to not only measure human uncertainty during collaborative tasks with robots in runtime, as well as to use such knowledge to adapt accordingly.

## 3. Use Case and Data Collection

### 3.1. Use Case: The FEROX Project

FEROX https://ferox.fbk.eu/ (accessed on 10 February 2023) is a project that aims to support workers collecting wild berries and mushrooms in wild and remote areas of Nordic countries by using robotic technologies. One of the key aspects of the project is its focus on HRC by deploying unmanned aerial vehicles (UAV) to monitor and assist groups of workers during field operations. This improves workers safety in remote environments, where access to help or assistance may be limited. The expected end results will be an increased worker trust in collaborating with robots, leading to larger number of berries harvested, higher quality berries for consumers, more efficient picking times, new level of worker safety in remote environments, and reduced worker exhaustion levels. Figure 1 depicts a view of the work field of the FEROX Project.

To achieve its aim, the FEROX project is exploring the use of wearable technology to infer the needs and states of the workers. One possible solution is to use a wearable device with integrated IMU (i.e., accelerometer, gyroscope and magnetometer) that can enable the identification of different activities, such as walking, running, sitting, collecting, and loading berries. Additionally, data from a global navigation satellite system (GNSS) (e.g., from the worker smartphone) can be used to infer activities performed over distance, such as driving a vehicle. It is foreseen that the combination of these two commonly adopted cheap devices would allow for more accurate and real-time monitoring of the workers’ activities and needs, enabling the project to better support and assist them.

Figure 2 illustrates the conceptual overview of the architecture aimed to be implemented in the FEROX Project. As stated above, human workers are equipped with wearables and other technologies, which feed the herein proposed FS-LSTM architecture to assess their behaviors and the associated uncertainty for a high-level decision-making system. The system may integrate human physiological and kinematic data to identify human activities, such as (1) human locomotive activities, including idle and walking; (2) human work-related activities, such as berry picking; (3) potential detection of human injury, combining physiological data, such as heart rate, in the future; and (4) a multi-UAV system that provides assistance to the human workers based on the output of the high-level decision-making system, which takes into account the human state defined by FS-LSTM. The next phase of the study will focus on developing the high-level decision-making system to explore the areas, track human location, and assist with loading the collected berries to the collection point (see Section 6).

### 3.2. Data Collection: FEROX Simulator and Synthetic Data

In recent years, the performance of HAR systems has seen significant advancement due to the use of deep learning techniques. However, the acquisition and labelling of large datasets for training and evaluating these methods can be time-consuming and costly. To address these limitations, one solution is to use synthetic datasets that do not require manual labelling or expensive hardware for data capturing [51]. This approach has several advantages, such as producing labelled data without human input, being beneficial in fields where data acquisition is costly, such as field robotics [52].

As a preliminary study, we present a simulator that generates automatically labelled synthetic data by simulating a human character with a chest-worn virtual IMU and smartphone GNSS sensors. The first goal of this study is to develop a simulation environment that can produce synthetic human motion data to feed a HAR system capable of recognizing different locomotive actions. The focus of this research is on developing a system that can be trained using only synthetic labelled data, and then tested and evaluated with real data to justify its reliability for further studies.

#### 3.2.1. FEROX Simulator Development and Virtual Sensor Modelling

We have developed the simulator using the Unity (https://unity.com/ (accessed on 10 February 2023)) game engine with the ultimate goal of creating a game-like environment for HRC. To achieve this, we initially focused on setting up the forestry scenario by using the Unity Terrain High-Definition Render Pipeline (https://assetstore.unity.com/packages/3d/environments/unity-terrain-hdrp-demo-scene-213198 (accessed on 10 February 2023)) and the avatar using the Mixamo library (https://www.mixamo.com/ (accessed on 10 February 2023)), contemplating simple actions, such as idle, walking, running, sitting, falling down and getting up as shown in Figure 3. We also implemented work-related actions, such as collecting and loading berries, driving a vehicle, etc. To generate the animations, we used a keyframe-based method that models connected virtual human body joints in a sequence of frames.

To establish communication between the different agents in the simulation and the developed framework, we integrated the ROS TCP Connector (https://github.com/Unity-Technologies/ROS-TCP-Connector (accessed on 10 February 2023)) to set up a TCP connection between Unity and the widely popular Robot Operating System (ROS) framework [53]. This allows us to generate C# classes to serialize and deserialize ROS messages, specifically the synthetic data obtained by the virtual IMU and GNSS sensors. The GNSS coordinate data is published at a rate of 5 Hz to the ROS network as a sensor_msgs/NavSatFix standard message. The synthetic IMU data is published at 50 Hz to the ROS network as a sensor_msgs/Imu standard message that stores the data over. Additionally, we also publish the current activity (label) being performed at 50 Hz to the ROS network under the message type std_msgs/String. All these three types of messages include a timestamp, which ensures that the activity labels can be synchronized with a given GNSS and IMU data stream.

In this study, we implemented a virtual model of the RION AH200C IMU sensor (http://en.rion-tech.net/products_detail/productId=158.html, (accessed on 10 February 2023)) which integrates an accelerometer, a gyroscope and a magnetometer, thus combining them and providing readings of linear acceleration, angular velocity and orientation. It is possible to place a virtual sensor in any desired position, as long as it is attached to a human joint. In our case, the virtual IMU sensor was placed on the chest of the avatar, as it is shown with a blue mark in Figure 3. The linear acceleration was calculated by taking into account the discrete derivative of the velocity with respect to the time as shown in Equation (Equation 1):(1)$$\begin{array}{c} a\left(t_{n}\right)=K\left(\frac{v\left(t_{n}\right)-v\left(t_{n-1}\right)}{T_{s}}\right) \end{array}$$
where *a*, *K*, *v*, and *T* stands for the linear acceleration, gain factor, velocity and time of the cycle, respectively. More particularly, we captured the position of the virtual IMU attached to the chest joint of the avatar and calculated its second discrete derivative every 20 ms on its own frame. In order to do that, we made use of the parent-child concept of Unity, which relies on a hierarchical structure between transform frames (position and rotation), as the pose of the child changes accordingly to the pose of the parent. The position of the child is applied from the current position of the avatar’s chest. The rotation of the parent is adopted from the current rotation of the avatar’s chest, while the position of the parent is adopted from the avatar’s chest position in the previous frame as illustrated in the Figure 4 in which the transparent human figure represents the virtual IMU position of the previous cycle. This concept allows us to obtain the position of the virtual IMU in its local frame.

It is noteworthy that, due to the successive discrete derivative, the linear acceleration is greatly affected by noise, which is not observable in the data retrieved from the real IMU sensor. Therefore, we have applied a linear interpolation followed by smoothing the data using an exponential smoothing algorithm commonly employed in time-series data to remove high-frequency noises, as in Equation (Equation 2), where $x_{t}$ is the data sequence, $s_{t}$ is the output of the exponential smoothing algorithm, *t* is time and α is the smoothing factor:(2)$$\begin{array}{cccc} \; & s_{t}=x_{t}, & t=0\\ \; & s_{t}=\alpha x_{t}+\left(1-\alpha \right)s_{t-1}, & t>0 & \;\mathrm{and}\;0<\alpha \leq 1 \end{array}$$

Additionally, and because real-world accelerometers are generally affected by gravitational acceleration, we have calculated the gravity vector in the local frame of the sensor by making use of the Unity physics engine.

In order to have the orientation information, we extracted the virtual sensor’s rotation in quaternions. Quaternions provide a convenient mathematical notation for representing the orientation of objects in space, being represented with complex numbers in the following form as shown in Equation (Equation 3), where $q_{x}$, $q_{y}$, $q_{z}$ are the vector units and $q_{w}$ is the scalar unit. Then, similarly to acceleration, we applied smoothing algorithm to smooth the quaternion data.
(3)$$q=q_{x}+q_{y}+q_{z}+q_{w}$$

At last, to obtain the angular velocity, we made use of the orientation described above, converting quaternions to Euler angles, and applying the related discrete derivative at every 20 ms, as represented in Equation (Equation 4), where ω, θ, and *t* represent angular velocity, rotation angle in radians, and time, respectively.
(4)$$\begin{array}{c} \omega =\frac{\Delta \theta }{\Delta t} \end{array}$$

In line with the approach taken to model the virtual IMU, a similar methodology was adopted to model GNSS data. Specifically, a smartphone positioning system was utilized as a reference for modelling GNSS data. To simulate the GNSS data, the avatar’s position in space was leveraged and converted into longitudinal and latitudinal coordinates using the GpsConverter package (https://github.com/MichaelTaylor3D/UnityGPSConverter (accessed on 10 February 2023)). It should be noted that the smartphone GNSS data had already undergone filtering, thus negating the need for additional data smoothing techniques [54].

While the data generated closely matches its real counterpart, real sensors are often affected by noise. Preliminary tests using the data generated from the aforementioned approach led to the overfitting of the models. The real IMU and GNSS sensors have noise characteristics due to some calibration errors or environmental noise that affects the sensor readings. Therefore, in order to make the synthetic data more realistic, a Gaussian Noise was injected on both sensors, more specifically affecting the longitude, latitude, linear acceleration and angular velocity variables. Noise was not added to quaternion data as the data provided by the real IMU sensor already comes from Extended Kalman Filter, which leads to a noiseless signal [55]. To add variability to the virtual IMU sensor data, the velocity and sequence of movements in the virtual avatar were adjusted in runtime. Different velocity levels can result in different patterns in the sensor data, while different sequences of activities can affect the overall variability of the data.

#### 3.2.2. Data Preparation

To justify the reliability of the synthetic data, we conducted a preliminary study by using MatLab to deploy the sequence classifier for training and testing, benefiting from both Deep Learning Toolbox (https://www.mathworks.com/products/deep-learning.html (accessed on 10 February 2023)) (for sequence data classification) and ROS Toolbox (https://www.mathworks.com/products/ros.html (accessed on 10 February 2023)) (for seemless communication with the ROS master).

We have started by building our own dataset, containing the synthetic data obtained by the virtual IMU and GNSS sensors, as well as the real data obtained by the real RION AH200C IMU sensor and GNSS data of a smartphone. At this stage, the dataset included data from only four activities (Sit, Fall Down, Get Up and Idle) with automatic labelling being performed for the synthetic data as described in the previous section, and manual labeling being performed for the real data, which would be required to not only assess the feasibility of the virtual IMU and GNSS models, but also to validate and evaluate the classifier.

We have implemented a method for synchronizing the timestamps of IMU and GNSS sensors. For each timestamp of the label data, the closest GNSS and IMU timestamps are found. The GNSS data is associated with the IMU and label data over a short time of 10 timestamps. Then the GNSS route involving longitude (ϕ) and latitude (λ) has been converted to Cartesian *x* and *y* coordinates.

Taking into account the use of a single IMU and GNSS, be it virtual or real, we have considered a feature vector s(t) that includes linear acceleration ($a_{x}$, $a_{y}$, $a_{z}$), angular velocity ($\omega_{x}$, $\omega_{y}$, $\omega_{z}$), quaternion ($q_{x}$, $q_{y}$, $q_{z}$, $q_{w}$), *x* and *y* being represented as follows:(5)$$s\left(t\right)=\left[\begin{array}{cccccccccccc} a_{x} & a_{y} & a_{z} & \omega_{x} & \omega_{y} & \omega_{z} & q_{x} & q_{y} & q_{z} & q_{w} & x & y \end{array}\right]$$

To tackle this classification problem, we adopted a Long Short-Term Memory (LSTM) network, which is known to be state-of-the-art supervised method for sequence data classification. As previously stated, LSTM is an improved type of recursive neural network and, instead of having a single neural network layer, it has four interacting layers, namely, cell state layer, input gate layer, forget gate layer and output gate layer. This enables it with the ability to “remember” information for a certain period, enabling learning-term dependencies [49]. Further detailed information on LSTM structure can be found in Section 4.2.

#### 3.2.3. Synthetic Data Validation

We conducted a preliminary experiment to investigate if the synthetic data is as adequate as the real data. For real-world experiments, we have used a smartphone and the RION AH200C IMU sensor in a chest-worn sensor setup. For both real-world and virtual experiments, we have recorded 192 activities as 48 samples of each action, for up to 3 s, at 5 Hz and 50 Hz, respectively for GNSS and IMU data. We also created the categorical array that holds the labels corresponding to these actions. The LSTM network was trained with the synthetic data and subsequently tested with the real data. The adaptive moment estimation optimizer was adopted, with a maximum epoch of 200.

The results of these experiments are presented in Figure 5 with the confusion matrix depicting the accuracy of the experiment including the performance of each activity. A result of 84.9% indicates that although the model performs with acceptable accuracy, several Sit and Get Up actions were incorrectly classified as Idle. While the initial findings indicate that the model trained using synthetic data can accurately classify the four specified activities when presented with real-world data, the upcoming sections will delve deeper into the evaluation of HAR using synthetic data across a wider range of activities. Within the context of the FEROX project, and to propose a more encompassing architecture, more complex activities will be included, such as forestry-work-related ones. Therefore, due to the simulator feasibility for generating data, the next sections encompass data collection from 13 different activities and, likewise, a novel approach for HAR under uncertainty.

## 4. Fuzzy State Long-Short Term Memory (FS-LSTM)

The proposed FS-LSTM framework for HAR under uncertainty is presented in Figure 6. The framework is comprised of five blocks, labeled A, B, C, D, and E. Block A is responsible for collecting human-related data through multimodal sensors, hereby assessed using a chest-worn IMU and a GNSS smartphone positioning system. Block B processes the IMU and GNSS data, including linear acceleration, angular velocity, orientation from the IMU sensor, as well as longitude and latitude from the GNSS, which are published at 50 Hz and 5 Hz, respectively and as previously described in Section 3. This data is then transformed into linguistic labels for Motion and Tilt through a fuzzification process in Block C, as further described in Section 4.1. These fuzzified Motion and Tilt features serve as inputs for both Block D and E. In Block D, the fuzzified feature set is used as input in LSTM state machine learning, where multiple networks are trained for each state to be executed during runtime, including a recovery state called Lost. This process is further detailed in Section 4.2. In Block E, uncertainty is managed through defuzzification in a closed-loop. The fuzzified inputs are used in the classification network, which was established in the previous iteration, and the generated classification score is first fuzzified and then defuzzified into a crisp value to determine whether to progress to the next state or remain in the current one. The details of this process will be further explained in Section 4.3.

### 4.1. Fuzzification of Features

This section presents the design of the feature fuzzification process for converting IMU and GNSS data into linguistic variables that will serve as inputs in the proposed FS-LSTM model. This stage is marked as C block in Figure 6.

Fuzzy logic allows computer systems to mimic the human-like thinking and make decisions under uncertain and imprecise information. For instance, the subjective and ambitious statement “the food is good” is enough for a person to decide how much to tip. This way of handling uncertainty is important when the reliable exact information is not available.

The numeric data collected by the FEROX simulator, as described in Section 3, includes linear acceleration ($a_{x}$, $a_{y}$, $a_{z}$), angular velocity ($\omega_{x}$, $\omega_{y}$, $\omega_{z}$), and quaternion ($q_{x}$, $q_{y}$, $q_{z}$, $q_{w}$), as well as the *x* and *y* coordinates. The process of fuzzification involves converting these crisp numerical data from sensors (either synthetic or real) into linguistic variables for use in the proposed FS-LSTM model. In particular, we have selected Motion and Tilt variables for fuzzification. We obtained velocity information as crisp data inputs for Motion and y-axis Euler angle (or pitch) for Tilt. To calculate the velocity, we benefit from the synchronized GNSS and IMU data (previously addressed in Section 3) to first generate a smooth 2D Cartesian position, *x* and *y*. The initial velocity is calculated as the derivative of the position:(6)$$\begin{array}{cc} \; & V_{x_{0}}\left(t\right)=\frac{x(t)-x(t-1)}{\delta t},\\ \; & V_{y_{0}}\left(t\right)=\frac{y(t)-y(t-1)}{\delta t} \end{array}$$

At each time step, we then calculate the velocity along the *x* and *y* axis as:(7)$$\begin{array}{cc} \; & V_{x}\left(t\right)=V_{x_{0}}\left(t\right)+a_{x}\left(t\right)*t,\\ \; & V_{y}\left(t\right)=V_{y_{0}}\left(t\right)+a_{y}\left(t\right)*t \end{array}$$
where *t* is the elapsed time and $a_{x}$ and $a_{y}$ are the linear acceleration measurements along *x* and *y* axis, respectively, obtained after applying a rotation to the linear acceleration measurements provided by the IMU, so as to align the body frame with the world frame. Finally, *V* is calculated as the magnitude of the velocity vector:(8)$$V\left(t\right)=\sqrt{V_{x}{\left(t\right)}^{2}+V_{y}{\left(t\right)}^{2}}$$

To quantify Tilt, the y-axis Euler angle θ (pitch), which represents the tilt of the chest forward or backward was calculated. The extracted quaternion values were converted to Euler angles in degrees following the principle presented in Equation (Equation 9):(9)$$\begin{array}{cc} \phi & =atan2(2\left(q_{w}q_{x}+q_{y}q_{z}\right),1-2\left(q_{x}^{2}+q_{y}^{2}\right))\\ \theta & =arcsin(2\left(q_{w}q_{y}-q_{z}q_{x}\right))\\ \psi & =atan2(2\left(q_{w}q_{z}+q_{x}q_{y}\right),1-2\left(q_{y}^{2}+q_{z}^{2}\right)) \end{array}$$

The calculated *V* and θ are inputs utilized in the fuzzification process. In particular, *V* is mapped into five linguistic labels for Motion, while pitch (θ) is translated into five linguistic labels for Tilt. The numerical data for both Motion and Tilt were mapped into five linguistic variables by Gaussian membership function using MatLab’s Fuzzy Logic Toolbox as follows:(10)$$S_{U}\left(t\right)=\left\{\begin{array}{c} Motion\to \left\{S,L,M,H,E\right\}\\ Tilt\to \left\{MLB,LB,ST,LF,ELF\right\} \end{array}\right.$$
where *S*, *L*, *M*, *H* and *E* represent the linguistic variables Stopped, Low, Medium, High, and Extreme for Motion, respectively, while MLB, LB, ST, LF and ELF represent Medium Lean Back, Lean Back, Straight, Lean Front, and Extremely Lean Front, for Tilt, respectively. $S_{U}\left(t\right)$ represents the feature vector which then is used as input for the FS-LSTM method.

In Figure 7, the five linguistic variables for Motion and Tilt are displayed, alongside an activity sequence in which the human is first idle, then walks, then becomes idle again, and finally opens the bag and collects some berries.

The activity sequence in this short period is represented as: Idle, Walk, Idle, Open Bag, and Collect, which are shown in the figure in the ground truth section. Additionally, the figure shows the precise numerical values of the Velocity (*V*) and pitch (θ) as the inputs for Motion and Tilt, respectively. It can be seen from the figure that the chosen linguistic variables effectively depict the activities. For instance, one can easily recognize when the human is walking by noticing that the Motion value falls between Low and Medium. The Open Bag and Collect activities also showcase distinct characteristics, such as leaning back while opening the bag and leaning forward while collecting berries. Thus, these activities and their fuzzified representations are clearly illustrated and may be used as features to the HAR architecture.

### 4.2. State Machine Learning

This section presents the modelling of human activity by exploring the transitions between different activity states. These states may involve specific activity sequences or individual activities. As an example, the use case discussed in Section 3 is depicted through a state diagram in Block E of Figure 6. This particular use case models the sequential activities of a human picker, including locomotive movements (such as Idle, Walk, Run), berry collection and loading, and vehicle driving. The flow is represented by 14 states (with the recovery state Lost) and includes states that occur in a sequential manner, such as sitting and driving a car, and transitions that occur between these states, such as Sit Down to Sit, Sit to Stand Up or Enter Car to Drive. There are also states that lead to multiple possibilities, such as the Idle state, which can transition to activities such as Open Bag, Walk, Run, Enter Car, or Sit Down.

The state transitions in a given process are modelled individually through the use of LSTM networks. As previously mentioned, LSTM is capable of extracting hidden patterns from long-term sequential data by handling gradient exploding or vanishing gradients problems [56]. In more detail, the internal structure of an LSTM network consists of multiple gates, including input gate $i_{t}$, forget gate $f_{t}$, and output gate $o_{t}$, that control the flow of information towards the final output. The input gate updates the information, while the forget gate processes information from both the input gate $i_{t}$ and the previous state $C_{t-1}$, selectively removing information from the current state $C_{t}$ when necessary. The output gate forwards the final output to the next LSTM unit and retains the output value for subsequent sequence predictions. The recurrent unit, on the other hand, estimates the state of the previous cell $C_{t-1}$ and the current input $x_{t}$ using a tanh activation function. The value of $h_{t}$ can then be calculated as the scalar product of the output gate $o_{t}$ and the tanh of the $C_{t}$. The ultimate output is obtained by passing $h_{t}$ to a softmax classifier [57].

Each LSTM network within the proposed HAR architecture is designed to receive sequential data as input (fuzzy features addressed in the previous section) and generate output predictions for only the feasible transition states. The objective of this approach is to ensure that there are no impossible transitions between states. For example, in a use case where the activities include Sit Down and Stand Up, or Collect and Load, it is possible that similar characteristics may result in an incorrect transition from one activity to the other. However, these transitions should not occur according to the expert-designed state flow, hereby represented as an FSM. The aim of this modelling approach is to guarantee that the predicted next state will be one of the possible states, providing important information for decision-making in an HRC system. Moreover, the number of possible classes significantly affects the size of the model, and a larger number of possible classes leads to a more complex model structure. Such a complex model demands additional computational resources, which would result in a longer runtime execution [58]. For instance, a Sit Down LSTM network with a reduced number of possible classes, such as Sit Down (remains in the same state) and Sit (moves to only the next possible state), is more efficient when compared to an LSTM network trained with all 13 possible activities, particularly when many of the outputs are unlikely to be the actual state.

The implementation of the LSTM layer in MatLab was carried out using the Deep Learning Toolbox. For the purpose of predicting the subsequent activity sequence, a sequence-to-sequence classification approach was employed. Each LSTM network corresponding to a specific state was trained using the same sequential input data. The input vector $S_{U}\left(t\right)$ comprises ten features, including the fuzzified values for Motion and Tilt, as described in Section 4.1 and depicted in Equation (Equation 11):(11)$$S_{U}\left(t\right)=\left[\begin{array}{cccccccccc} S & L & M & H & E & MLB & LB & ST & LF & ELF \end{array}\right]$$
where,
(12)$$\begin{array}{cccc} S & =\mathrm{Stopped} & MLB & =\mathrm{Medium}\;\mathrm{Lean}\;\mathrm{Back}\\ L & =\mathrm{Low} & LB & =\mathrm{Lean}\;\mathrm{Back}\\ M & =\mathrm{Medium} & ST & =\mathrm{Straight}\\ H & =\mathrm{High} & LF & =\mathrm{Lean}\;\mathrm{Front}\\ E & =\mathrm{Extreme} & ELF & =\mathrm{Extreme}\;\mathrm{Lean}\;\mathrm{Front} \end{array}$$

It is noteworthy, however, that each state-based LSTM network output sets differ, with each output set corresponding to a specific state, including the potential transitions, and including the possibility of remaining in the same state, as it is shown in Figure 8. Any non-feasible state names are labelled as Lost. The Lost network, which uses the same sequential input data and an output dataset of all states, serves as a recovery mechanism, being only activated when such class is output by the previous LSTM network.

The data generated in this study have been made publicly available (https://gitlab.ingeniarius.pt/ingeniarius_public/ferox/hrc-ferox.git (accessed on 22 February 2023)). The dataset includes the following features, as shown in Equation (Equation 13): $t_{s}$ is the timestamp, s(t) is the feature vector containing the raw sensor data shown in Equation (Equation 5), $s_{U}\left(t\right)$ is the feature vector containing the fuzzified sensor data shown in Equation (Equation 11), and *G* is the ground truth of activities presented as a string array where each element represents the activity label at a specific timestamp.
(13)$$\left[\begin{array}{cccc} t_{s} & s(t) & S_{U}\left(t\right) & G \end{array}\right]$$
(14)G=[“Idle”,“Idle”,“Sit”,…,“CloseBag”]

### 4.3. Coping with the Uncertainty through Defuzzification

The trained network models described in the preceding section are employed in a closed-loop architecture to predict the subsequent state, as depicted in Block E of Figure 6. The fuzzified outputs generated by Block C serve as inputs to the trained network model associated with the current state, which is referred to as the Lost network in the first iteration and is updated in subsequent iterations in accordance with the flow of the architecture.

The proposed architecture makes use of network model classification scores as posterior probabilities based on the fuzzified input set. These probabilities are calculated based on Bayes’ Theorem:(15)$$\hat{P}\left(B|A\right)=\frac{P(A|B)P(B)}{\sum_{j=1}^{R}P\left(A|j\right)P\left(j\right)}$$
where $\hat{P}\left(B|A\right)$ is the posterior probability that an observation *A* of given class *B*, P(A∣B) is the conditional probability of *A* given class *B*, P(B) is the prior probability for class B and *R* is the number of classes in the response variable [59]. The classification scores are represented as an *z*-by-*R* matrix, where *z* is the number of observations in the data and *R* is the number of unique classes. The matrix indicates the probability of each observation belonging to a specific class, with the predicted class being determined by the class with the highest score.

FS-LSTM falls on the assumption that state network models may struggle to make confident predictions if the highest score is low or if the scores are similar across classes, leading to a certain level of uncertainty. Yet, it uses such level of uncertainty to still generate a prediction based on the class with the highest score. It is therefore important to consider the level of confidence in the prediction and interpret the scores before making any decisions based on the model’s output. The uncertainty in the scores is evaluated using the fuzzy logic system.

Hence, similar as carried out for Motion and Tilt fuzzy variables, classification scores are used for the fuzzification to produce fuzzy linguistic labels for Uncertainty as Low, Medium, and High. The triangular fuzzifier is used to determine the degree of membership for each value. Unlike Motion and Tilt fuzzy variables, however, inference is followed by adopting a set of rules designed for each state, which are then later utilized to assess the uncertainty in the classification scores generated by the current state network. An example of fuzzy rules is presented for the Close Bag state. The uncertainty is assessed in the classification scores generated for the possible transition outputs of the Idle, Close Bag, and Lost recovery state by implementing these rules:IF Idle is Low and Close Bag is Low and Lost is Low THEN Uncertainty is HighIF Idle is Medium and Close Bag is Medium and Lost is Medium THEN Uncertainty is HighIF Idle is High and Close Bag is High and Lost is High THEN Uncertainty is HighIF Idle is High and Close Bag is Low and Lost is Low THEN Uncertainty is LowIF Idle is Low and Close Bag is High and Lost is Low THEN Uncertainty is LowIF Idle is Low and Close Bag is Low and Lost is High THEN Uncertainty is LowIF Idle is High and Close Bag is not Low and Lost is not Low THEN Uncertainty is HighIF Idle is not Low and Close Bag is High and Lost is not Low THEN Uncertainty is HighIF Idle is not Low and Close Bag is not Low and Lost is High THEN Uncertainty is High

In the fuzzy inference, the rules are applied to the fuzzified inputs to calculate the degree of fulfilment for each rule through aggregation. The following step is defuzzification, which transforms the fuzzy outputs into crisp outputs by using the fuzzy sets and their corresponding membership degrees. The result of the aggregation is converted into a crisp output value through the centroid method. The output of this system expresses the uncertainty as a crisp value between 0 and 1. This crisp value coming from the fuzzy logic system is then used to assess the confidence level of each network model before determining the next state. This is performed by comparing it to an experimentally-defined threshold. If the crisp value of uncertainty generated through defuzzification is lower than the threshold, this implies the model is confident in the prediction and the classification is carried out based on the highest score, which is identified as the next state. This state could be a different state or remain unchanged. The network model that corresponds to the predicted state is then selected for use in the next iteration. If the level of uncertainty exceeds the established threshold, the system remains in the same state. In this case, the next iteration performs the classification utilizing the current network model. This iterative process continues in accordance with the closed-loop architecture until the model reaches a sufficient level of confidence in its prediction.

Figure 9 shows a small section of the timeline of activity recognition handling the uncertainty. Between 62 and 65 samples, the model experiences low confidence in its predictions as the uncertainty is above the threshold and remains in the same state for subsequent selections. At sample 184, the model erroneously classifies the data as Walk while the ground truth remains Run. While it may seem like a premature classification, it could be a coincidence since transitions from Run to Walk are possible. However, at sample 185, it is observed that the Walk state network model lacks certainty in the classification, causing it to remain in the Walk state. The uncertainty level drops below the threshold once the ground truth and the predicted classes align at sample 186.

## 5. Results and Discussion

### 5.1. Benchmark of Activity Recognition Performance

In this section, we present the results of our experimental study designed to evaluate the effectiveness of the proposed approach. We used a training dataset consisting of 13 different activities (Idle, Walk, Run, Sit Down, Sit, Stand Up, Exit Car, Drive, Enter Car, Open Bag, Collect, Load, Close Bag) as depicted in Figure 6 block E. The training dataset was collected over a period of 17.85 min and includes 539 activities, with a total of 53,553 samples of virtual IMU data and 5355 samples of virtual GNSS data. The testing dataset consists of 168 activities and comprises 21,188 samples of virtual IMU data and 2118 samples of virtual GNSS data, spanning a total duration of 7.06 min. In this section, the number of hidden layers was set to 64 for all LSTM networks considered.

We have benchmarked three different methodologies:

(a) Traditional LSTM: a LSTM model that was trained using raw sensor input, similar to what has been presented in Section 3, though with all 13 states instead of four.

(b) Fuzzy LSTM: a LSTM model that was trained with the fuzzified features described in Section 4.1 and outputs all 13 states.

(c) FS-LSTM: the multiple LSTM models that were trained for each state, using fuzzified features, and each model only outputs the feasible states, as per presented in Figure 6 and described in the previous Section 4.

For the Traditional LSTM model, the input feature vector s(t) is the one previously presented in Equation (Equation 5), consisting of linear acceleration values ($a_{x}$, $a_{y}$, $a_{z}$), angular velocity values ($\omega_{x}$, $\omega_{y}$, $\omega_{z}$), quaternion values ($q_{x}$, $q_{y}$, $q_{z}$, $q_{w}$), and the *x* and *y* Cartesian coordinates. The input vector for both the Fuzzy LSTM and FS-LSTM models, $S_{U}\left(t\right)$, includes the five linguistic labels previously described in Section 4.1 for each of the extracted Motion and Tilt variables (Equation (Equation 11)).

The results of the three methodologies are presented in Figure 10 as confusion matrices, which depict the outcomes of (i) Traditional LSTM; (ii) Fuzzy-LSTM; and (iii) FS-LSTM. The rows represent the target classes, while the columns represent the output classes. Superior classification accuracy results are identified in bold. The results indicate that utilizing solely raw sensor data leads to a significantly low accuracy of 23.2%. Conversely, by utilizing fuzzified inputs, the Fuzzy-LSTM approach markedly enhances the accuracy to 93.2%. This demonstrates how a fuzzy logic system handles data ambiguity and achieves correct classification, where the Traditional LSTM method frequently falls short. The proposed FS-LSTM methodology achieved an accuracy of 90.9%. This result was obtained by treating samples classified as Lost as unchanged. This approach ensures that the system waits until it recovers from the Lost state, which runs the same model as the Fuzzy-LSTM approach (with 13 state outputs), before transmitting the predicted output to a higher-level decision-making system to ensure the correctness of the transmitted output.

Although the overall accuracy does not show a significant difference from the Fuzzy-LSTM, being even slightly inferior in terms of accuracy, the FS-LSTM prevents transitions that should not occur from happening. This might result in slightly superior performance on Sit Down-Sit-Stand Up transitional states when compared to Fuzzy-LSTM. This not only avoids passing incorrect information to the higher-level management system but also improves the probability of predicting the next state. Figure 11 illustrates this prevention of wrong transitions more clearly for one of the many sequences generated by the aforementioned models. In the figure, a sequence of activities is given with their ground truth in the blue dotted line and predicted outputs with the straight black line obtained through Traditional LSTM (top), Fuzzy-LSTM (middle), and FS-LSTM (bottom). Once again, it is shown that Traditional LSTM is unable to classify activities correctly, alternating between Walk and Idle in this sequence. For the Fuzzy-LSTM, around sample 150, the predicted class is Enter Car, while the ground truth class is Idle. In contrast, in the same sample in FS-LSTM, the classification of the current network model (Idle state network) is either correct, or the uncertainty is high, and the system remains unchanged in Idle. This prevents the system from making infeasible transitions to Enter Car, which would break the sequence of predicted activities. Fuzzy-LSTM, however, makes infeasible transitions multiple times, such as in sample 665, where it transitions from Idle to Close Bag, which should never happen. FS-LSTM handles such infeasible transitions through uncertainty defuzzification and the state machine approach generally well. As it is described earlier, FS-LSTM handles misclassifications via recovery of the Lost state. In the confusion matrix Figure 10, the Lost classified outputs are not shown as they were treated as unchanged states. However, in practice, when the network is unable to classify a sample in any of the possible transitions, the decision-making moves to the Lost state. For instance, in the sample around 956 marked with a red circle, the system is in the Lost state. This occurred because the Idle network predicted Enter Car instead of Sit Down, which was feasible but wrong. The Enter Car state network model was unable to classify the samples in any feasible transitions, and the system went to the Lost state before recovering to the Sit Down state.

In comparison, while Fuzzy-LSTM mostly performs poorly in the presented time window, it still achieves overall accurate classification, as demonstrated by the corresponding confusion matrix. However, it is important to note that Fuzzy-LSTM should be regarded as a sequence of activity flow prediction rather than individual sample prediction, underscoring the superiority of FS-LSTM. Furthermore, FS-LSTM is expected to achieve the same level of accuracy, of even higher, under constrained resources, since the multiple LSTM networks it encompasses are expected to not require the same number of hidden layers given the reduced number of outputs foreseen by each. This is further explored in the next section.

### 5.2. Benchmark of Efficiency

In this section, we evaluate the classification performance of the proposed FS-LSTM compared to Fuzzy-LSTM from the perspective of GPU resource efficiency. In the previous section, both models were trained with 64 hidden layers, and while Fuzzy-LSTM classified 13 states, FS-LSTM only used this complex network in the Lost state. To better understand the impact of this difference on computer resources, we monitored GPU utilization and power consumption during the classification runtime process for a period of 25 minutes. Our experiments were conducted using an NVIDIA GeForce GTX 1050. The benchmark chart for both models is presented in Figure 12 and Figure 13.

As shown in the last two rows of columns from Figure 12, despite both methods performing with high accuracy, Fuzzy-LSTM (represented by the dark blue bars) relies more heavily on GPU resources than the FS-LSTM approach (represented by the light blue bars). This is a critical consideration in any long-term outdoor application where computer resources are constrained. While in the last two rows of columns from Figure 13 the power consumption did not show any significant difference between these methods over a short-term test, high GPU load is a critical consideration in any long-term outdoor application where computer resources are constrained. A high GPU load indicates that the GPU is being heavily utilized to complete the classification task, which can cause the GPU to generate more heat and consume more power, ultimately affecting the overall performance and energy efficiency of the system over an extended period.

A question may arise as to whether reducing the number of hidden layers in both models can reduce its complexity, then leading to better efficiency, while still maintaining the desired accuracy. To answer this, we extended the benchmark study to five different combinations: both Fuzzy-LSTM and FS-LSTM networks trained with 32 layers (denoted with *), and a hybrid version of FS-LSTM (denoted with **) trained with 32 layers, except for the Lost network, which was trained with 64 layers. As shown in Figure 12 and Figure 13, reducing the number of layers significantly decreases GPU utilization and power consumption. Table 1 further compares the mean and standard deviations of GPU utilization and power consumption, demonstrating that a smaller model size with fewer output and hidden layers leads to significantly higher efficiency. However, reducing the number of hidden layers sacrifices accuracy, as shown in the confusion matrix in Figure 14.

These conclusions are not new, though what can be seen is that Fuzzy-LSTM* (i) and FS-LSTM* (ii), show a decrease in accuracy of 74.7% and 76.0%, with FS-LSTM slightly dethroning Fuzzy-LSTM under a lower number of resources (32 hidden layers instead of 64 hidden layers), while still requiring less GPU utilization. Furthermore, while the FS-LSTM 64 hidden layer Lost network (with 32 hidden layers for all other states), FS-LSTM** (iii), presents a similar GPU utilization and power consumption than the Fuzzy-LSTM with a 32 hidden layer (Fuzzy-LSTM*), its accuracy rises to 81.7% (versus the 74.7% of Fuzzy-LSTM*). While this is not an outstanding result, it can be a useful compromise between model size, performance, and energy efficiency in certain applications. One such application is covered by the FEROX Project, where the system is expected to run on smaller portable devices, such as smartphones and wearables. In such scenarios, an efficient utilization of computing resources becomes crucial, and the FS-LSTM model may offer a viable solution without significantly reducing the human activity recognition accuracy.

### 5.3. Discussion

As described in Section 1.2, this paper presented three key incremental developments, which have been successfully achieved and summarized as it follows:Enhance Long Short-Term Memory (LSTM) networks by incorporating fuzzy logic to model human uncertainty (Fuzzy-LSTM): the objective of this research was to enhance the accuracy of LSTM networks by incorporating fuzzy logic to model human uncertainty. Even though the preliminary results, as shown in Figure 5, depicted an accuracy of 84.9% for recognizing four activities, such accuracy dropped to 23.3% when trying to recognize 13 different activities. By utilizing fuzzified Motion and Tilt features, the Fuzzy-LSTM model was capable to effectively handle uncertain data. The initial results of the study showed that Fuzzy-LSTM improved the accuracy of activity recognition by a significant margin, achieving 93.2% accuracy compared to the traditional LSTM model using raw sensor data.Extend the Fuzzy-LSTM approach by incorporating finite-state machines (FSM) to model activity sequences, resulting in the Fuzzy State LSTM (FS-LSTM) model: the primary objective of this research was to improve the predictability of human activity sequences by identifying possible transitions between states. While Fuzzy-LSTM achieved 93.2% accuracy under unconstrained GPU resources, approximately 3% more than FS-LSTM, it often resulted in infeasible transitions. However, when using constrained resources such as embedded systems or limited GPU resources, the FS-LSTM showed superior performance compared to Fuzzy-LSTM. As shown in Figure 14, FS-LSTM has a trade-off between accuracy and computational resources, but it offers significant benefits for long-term real-time outdoor applications.Develop a defuzzification-based method to estimate human uncertainty by aggregating predicted scores of the LSTM model: this proposed approach aimed to estimate the uncertainty associated with the LSTM classifier’s predictions through defuzzification. By waiting until the prediction became certain, the system could achieve an accuracy of 90.0%. This development is crucial to prevent the system from making wrong transitions between states before the model becomes certain, thereby improving the overall performance of the system. Although this study did not show the direct impact of this issue, it could significantly affect the high-level decision-making process for robots, where the system needs to consider the current human state. Any infeasible transitions could compromise the performance of the system, causing trust and safety issues.

This study presents a few drawbacks and limitations that should be taken into consideration. Firstly, the results were obtained using only synthetic data. Although the preliminary results (Figure 5) show that models trained with such data are transferable to real-world data, this may still not accurately reflect real-world scenarios or domains without further modification and customization. Therefore, caution should be exercised when applying the proposed approach to other contexts. Furthermore, while synthetic data provides advantages such as low-cost data generation and easy labelling, it may not fully capture the complexity and variability of real-world data. Thus, a further study is needed to ensure the generation of data with the same characteristics as real data when using a different type of sensor. Additionally, in a real-world setup, the sensor placement and other aspects of the experimental setup may need to be adjusted to account for different environmental conditions and potential interferences.

## 6. Conclusions and Future Work

This study presents an architecture for human activity recognition and modelling intended for use in human-robot collaboration in field applications. The approach employs multiple LSTM networks, each trained to recognize feasible states within an FSM architecture. The FSM architecture is further enhanced with fuzzy logic to determine the uncertainty level of the classification made by the LSTM, thereby preventing unfeasible activity transitions in high-level decision-making systems. The proposed approach is compared to a traditional LSTM model trained on raw sensory data and a Fuzzy-LSTM model that used fuzzified sensory data as inputs to train a single LSTM network.

The proposed approach achieves high accuracy, with a rate of 90.9%, while efficiently utilizing computer resources. The system’s performance is evaluated using synthetic data generated from a berry collection use case developed in a simulator. Future work will involve assessing the system’s performance using real-world data within the FEROX project, as well as optimizing the developed work to operate on a small platform such as a smartphone.

As a continuation of this work, the next step will involve developing a high-level decision-making system that utilizes the human state predicted by the proposed FS-LSTM, as well as its associated uncertainty, to make informed decisions for each agent in a multi-robot and multi-human system. The decision-making system will be developed based on explicit and implicit relationships between agents, building upon the presented study’s understanding of human activity.

## Figures and Tables

**Figure 1 sensors-23-03388-f001:**
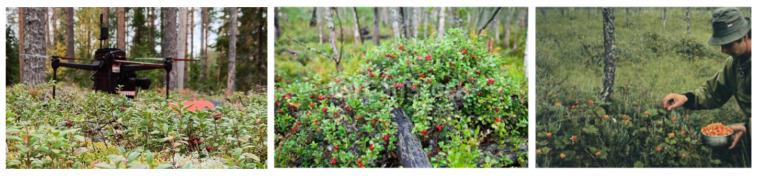
A view on the work field of the FEROX Project.

**Figure 2 sensors-23-03388-f002:**
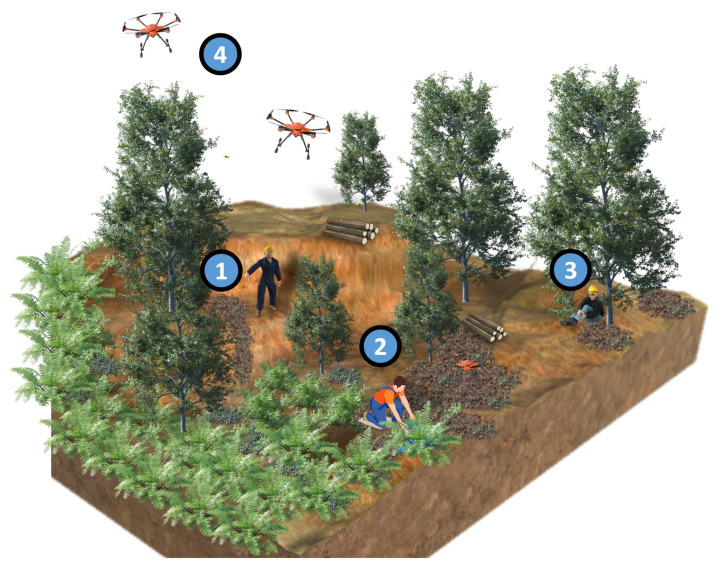
An overview of the use case scenario.

**Figure 3 sensors-23-03388-f003:**
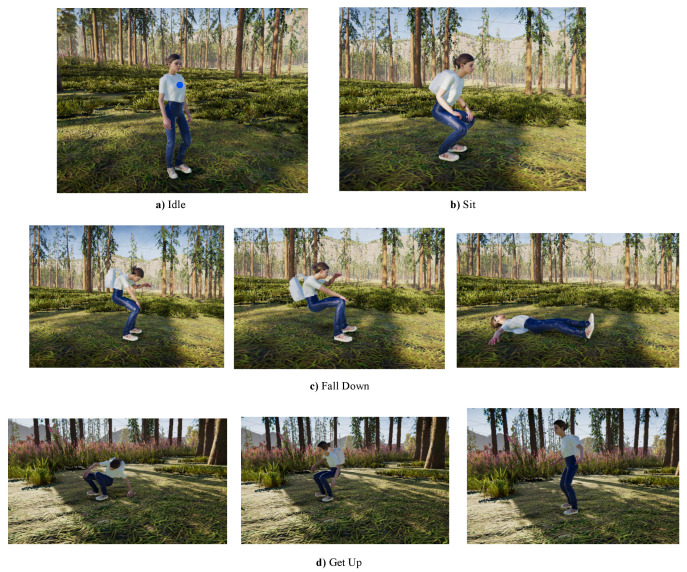
The avatar performs locomotive actions.

**Figure 4 sensors-23-03388-f004:**
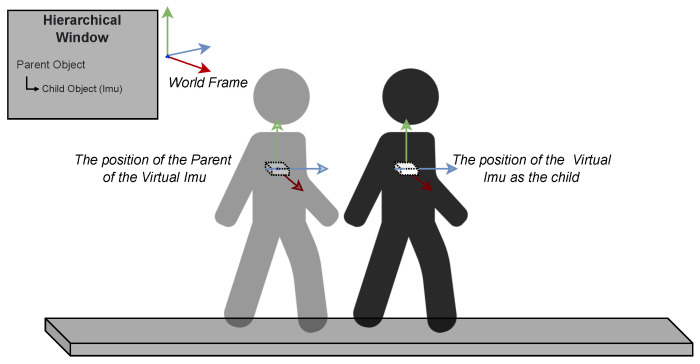
The parent-child relationship is adopted to obtain the position data in virtual IMU’s local frame.

**Figure 5 sensors-23-03388-f005:**
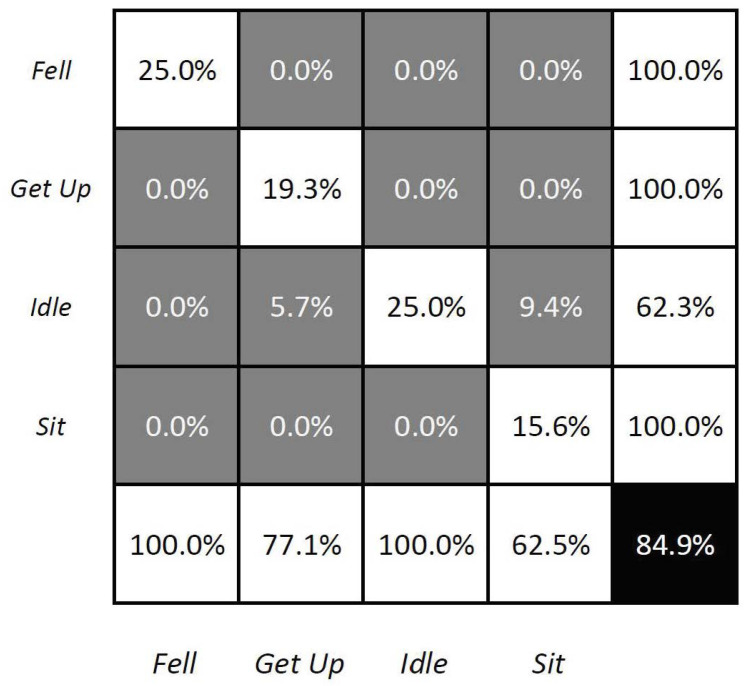
The confusion matrix of the LSTM network.

**Figure 6 sensors-23-03388-f006:**
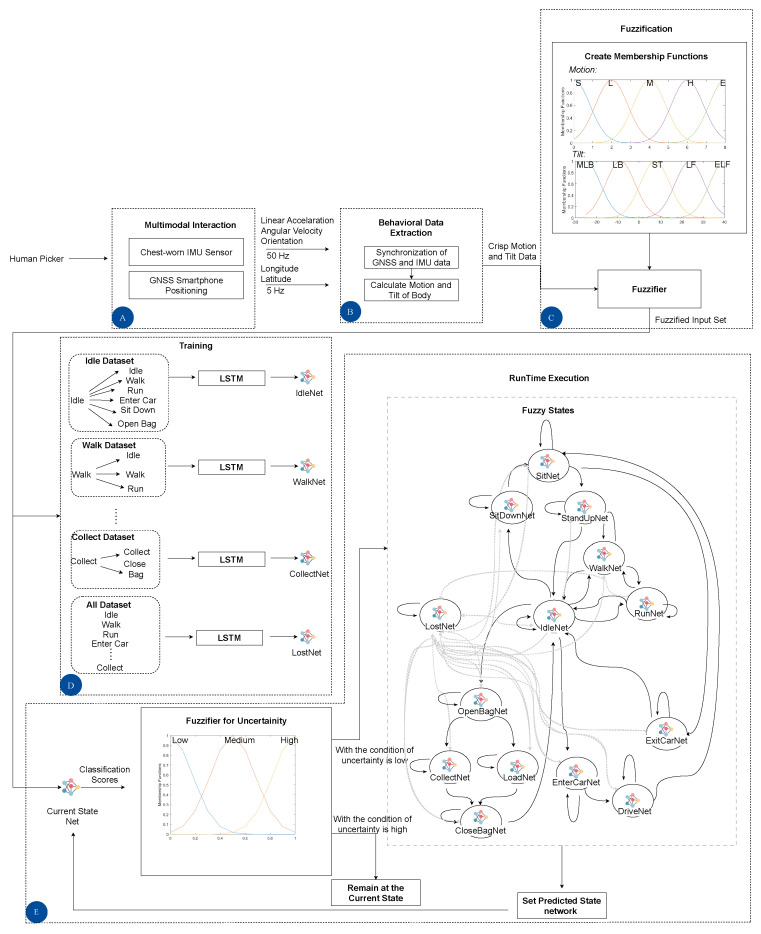
The diagram of the proposed architecture.

**Figure 7 sensors-23-03388-f007:**
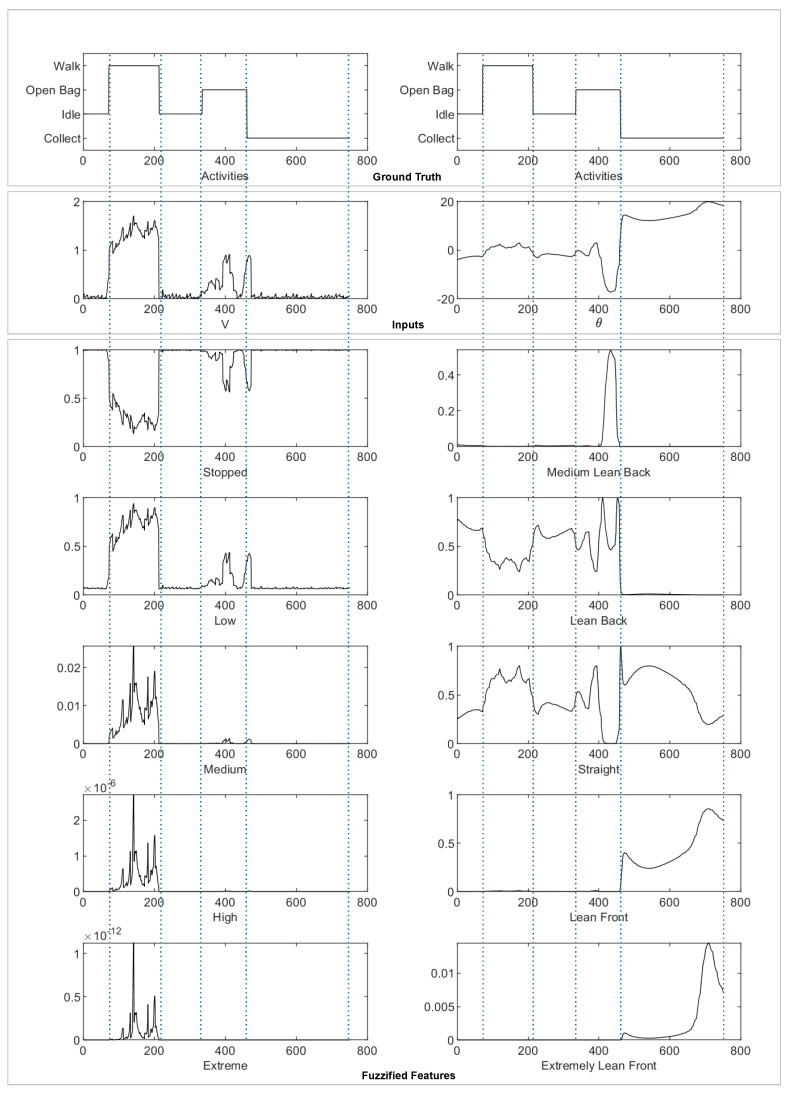
Motion and Tilt plotting along an activity sequence.

**Figure 8 sensors-23-03388-f008:**
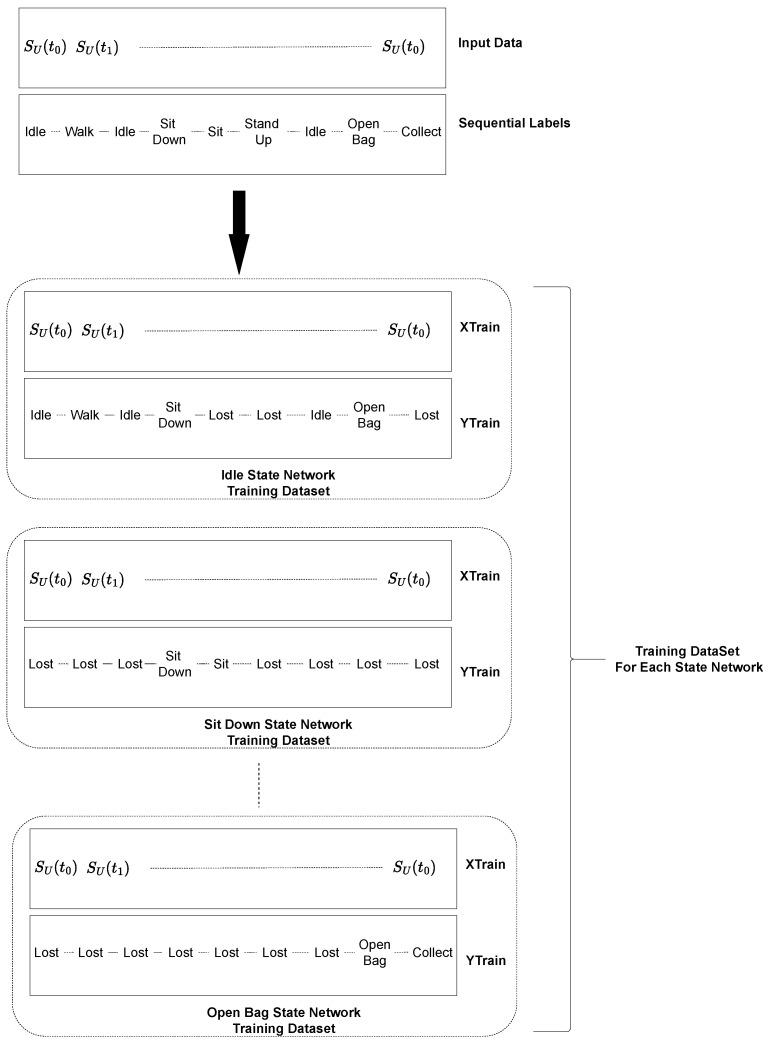
LSTM training datasets.

**Figure 9 sensors-23-03388-f009:**
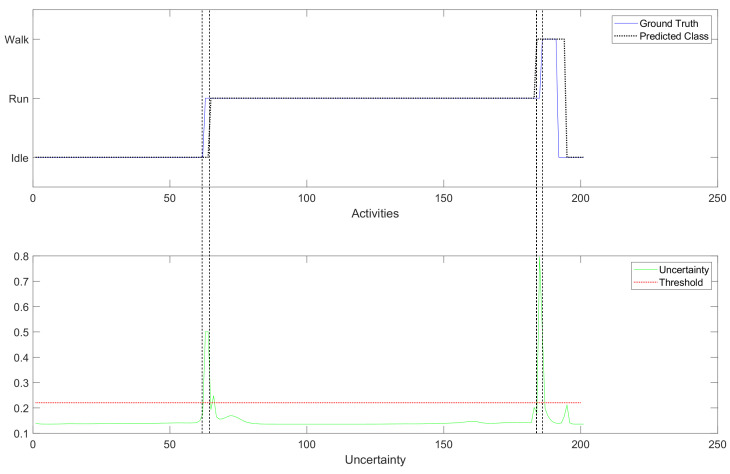
Accessing uncertainty in the activity recognition.

**Figure 10 sensors-23-03388-f010:**
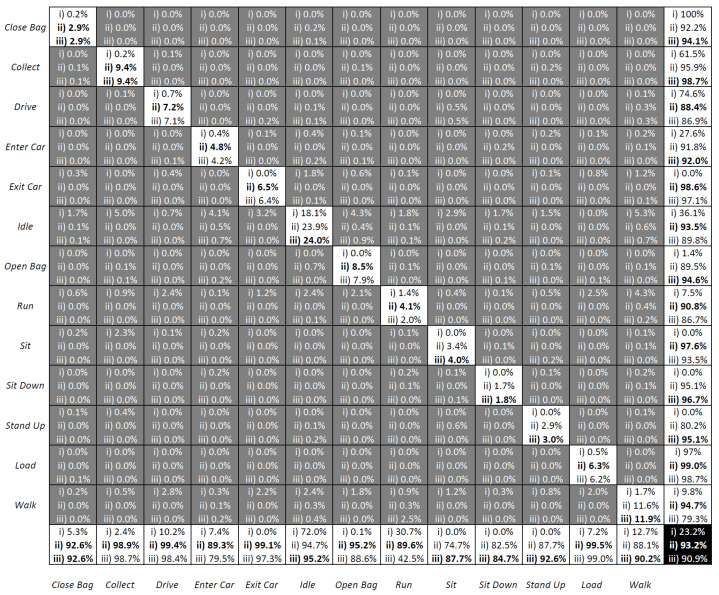
The confusion matrix of Traditional LSTM, Fuzzy-LSTM and FS-LSTM. Superior classification accuracy results are identified in bold.

**Figure 11 sensors-23-03388-f011:**
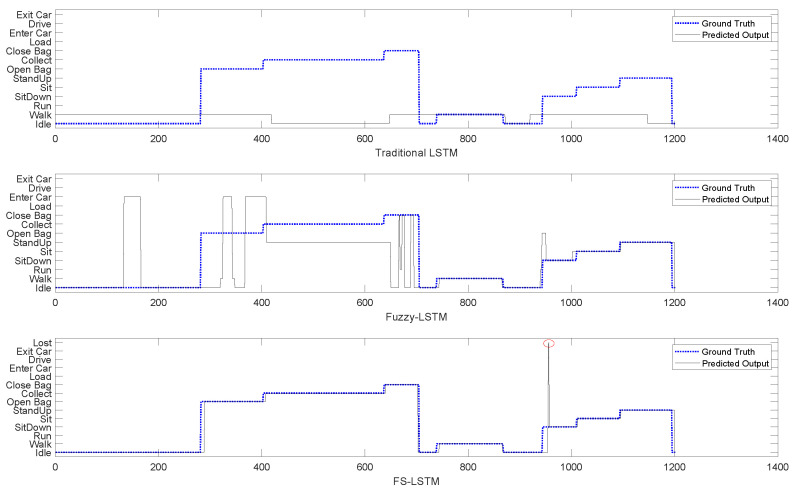
The benchmark of three predicted output sequences via Traditional LSTM, Fuzzy-LSTM and FS-LSTM. The Lost state is marked with the red circle.

**Figure 12 sensors-23-03388-f012:**
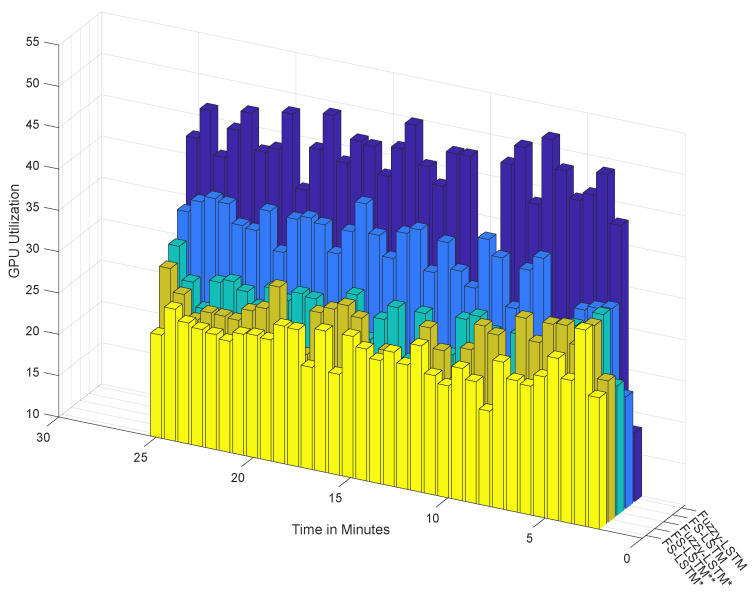
The benchmark of GPU utilization during testing. Fuzzy-LSTM and FS-LSTM networks trained with 32 layers (denoted with *), and a hybrid version of FS-LSTM (denoted with **).

**Figure 13 sensors-23-03388-f013:**
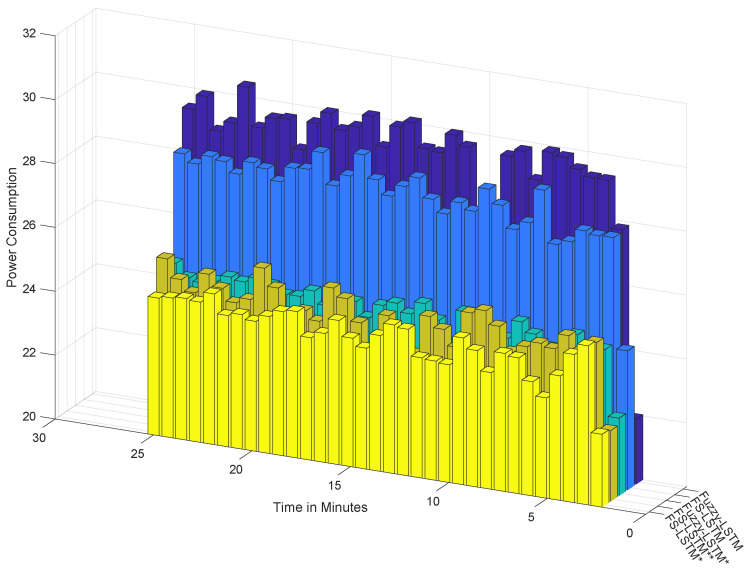
The benchmark of power consumption during testing. Fuzzy-LSTM and FS-LSTM networks trained with 32 layers (denoted with *), and a hybrid version of FS-LSTM (denoted with **).

**Figure 14 sensors-23-03388-f014:**
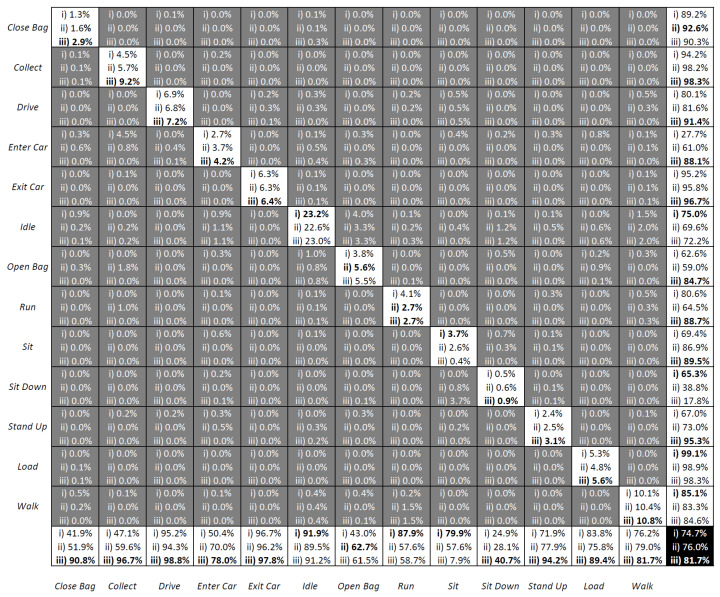
The confusion matrix of Fuzzy-LSTM* (i), FS-LSTM* (ii) and FS-LSTM** (iii). Superior classification accuracy results are identified in bold.

**Table 1 sensors-23-03388-t001:** The mean ± standard deviation of GPU utilization and power consumption. Fuzzy-LSTM and FS-LSTM networks trained with 32 layers (denoted with *), and a hybrid version of FS-LSTM (denoted with **).

Mean ± SD	Fuzzy-LSTM	FS-LSTM	Fuzzy-LSTM *	FS-LSTM *	FS-LSTM **
GPU Utilization	44.55 ± 9.85	35.21 ± 9.57	28.05 ± 9.56	25.58 ± 9.46	27.54 ± 10.16
Power Consumption	29.25 ± 2.04	28.14 ± 1.91	24.66 ± 0.89	24.25 ± 0.89	24.70 ± 1.34

## Data Availability

The data generated in this study are openly available here: https://gitlab.ingeniarius.pt/ingeniarius_public/ferox/hrc-ferox.git (accessed on 22 February 2023)).

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
