# Peer review of "Human-Aware Collaborative Robots in the Wild: Coping with Uncertainty in Activity Recognition"

_sensors, 2023, doi:10.3390/s23073388_

Round 1
Reviewer 1 Report
The work is interesting, and the results are convincing according to the statement by the authors. According to my understanding, the work
- Introducing a novel approach, called Fuzzy State-Long Short-Term Memory (FS-LSTM), to address the challenge of human behavior uncertainty during Human-Robot Collaboration in dynamic and unstructured environments.
- Developing a combined activity recognition and sequence modeling system using state machines and LSTM deep learning method that fuzzifies ambiguous sensory data for better representation of high-level human behavior based on low-level input data.
- Comparing the proposed FS-LSTM approach with traditional LSTM approaches with raw sensory inputs or fuzzy inputs. The results show significant improvement in accuracy when using fuzzified inputs compared to traditional methods.
- Proposing an evaluation process that includes quantifying prediction uncertainties within the FSMs used by FS-LSTMs. This allows accepting or rejecting transitions between states based on certainty levels which can be useful for future work in HRC where certainty is important.
-It does not mention any specific dataset used for the experiments. However, it describes a use case scenario of workers collecting wild berries and mushrooms in remote areas using unmanned aerial vehicles (UAVs) to monitor and assist them during field operations. The proposed approach is evaluated by comparing traditional LSTM with raw sensory data inputs, Fuzzy-LSTM with fuzzified inputs, and FS-LSTM approaches.
Reviewer 2 Report
In this research, the authors proposed a method to deal with human behaviour uncertainty during human-robot collaboration using deep learning methods. This research could benefit the human activity recognition area. However, some improvements are required as follows:
- The authors stated that this research investigates human activities in different fields, such as agriculture, forestry, and construction. However, only basic movements have been discussed, such as getting up sit shown in Figure 4. Although Figure 9 demonstrated more different complicated activities, more discussions and evaluations are needed to support their stated contributions.
- Did the authors use data directly from the FEROX project? The authors may provide a dataset list that was used in this research as a table to help readers to understand.
- Please provide IMU accuracy detailed information such as errors and accuracy.
- Because authors use maxim 50Hz to analyze data and smooth them using Eq.2 based on the linear method, how did authors define alpha that can ensure no important nonlinear properties are deleted?
- In Figure 3 and Eq. 4, Which joints are used to compute velocity? Is it the chest joint? Why did the authors not choose the hip joint, which is the root joint of the human body's hierarchal structure?
- Figures texts are too small to see, such as Figure 10.
- The lSTM is not a state-of-the-art model among the deep learning methods. How about a transformer? Did the authors test their data with a transformer model?
Reviewer 3 Report
The study proposes a new approach called Fuzzy State-Long Short-Term Memory (FS-LSTM) to deal with human behavior uncertainty in dynamic and unstructured environments.The proposed approach addresses this by fuzzifying the sensory data and developing a combined activity recognition and sequence modelling system using state machines and the LSTM deep learning method. The evaluation shows that the use of fuzzified inputs significantly improves accuracy compared to traditional LSTM, and the fuzzy state machine approach offers the added benefits of ensuring feasible transitions between activities with improved computational efficiency.
First of all, I would like to congratulate the authors for their careful and fluent writing and article organization.
1- The study content is very intense. For a reader-friendly approach, it would be better to give the method overview or general flow first and then the rest of the presentation.
2- What are the limits, assumptions and drawbacks of the study? A benchmarking should be done with similar studies. What would change if it was done in real environment and with sensors? What advantages did data synthesis provide?
3- The contributions mentioned in the Introduction section should be handled and discussed under separate headings in the results section.
4- How the robot interaction takes place should be supported with a few visuals.
Round 2
Reviewer 2 Report
The authors have responded to the reviewer's concerns.
Reviewer 3 Report
Paper can be accepted in present form